# The Interplay among Empathy, Vicarious Trauma, and Burnout in Greek Mental Health Practitioners

**DOI:** 10.3390/ijerph20043503

**Published:** 2023-02-16

**Authors:** Kalliope Kounenou, Antonios Kalamatianos, Panagiota Nikoltsiou, Ntina Kourmousi

**Affiliations:** 1Department of Education, School of Pedagogical & Technological Education, 15122 Marousi, Greece; 2Department of Education, School of Education, University of Nicosia, Nicosia 2417, Cyprus; 3Department of Special Education, University of Thessaly, 38221 Volos, Greece

**Keywords:** burnout, vicarious trauma, empathy, mental health professionals

## Abstract

Background. Mental health professionals are at risk of experiencing vicarious trauma and burnout as a consequence of the nature of their work. Studies and scholars so far have demonstrated that empathy interacts directly with burnout, and they imply an interaction with vicarious trauma. However, research has paid little attention to the interplay among vicarious trauma, empathy, and burnout in mental health professionals who practice psychotherapy. This study examines the interplay between mental health professionals’ (those practicing psychotherapy) vicarious trauma and empathy and investigates the ways they contribute to burnout. Methods. The sample consisted of 214 mental health professionals (32 males and 182 females), working in the public and private sectors. Specific instruments were administered online to the sample: (a) an improvised demographic questionnaire (age, gender, education, specialty, years of experience, years of supervision); (b) the Counselor Burnout Inventory, validated for the Greek population by Kounenou et al.; (c) the Vicarious Trauma Scale; and (d) the Jefferson Scale of Physician Empathy. Results. Correlation analysis showed that empathy and vicarious trauma were positively related to burnout. Moreover, multiple regression analysis revealed that supervision, empathy, and, to a greater degree, vicarious trauma have a significant impact on burnout. Conclusion. Unlike relative research on burnout, gender and work experience did not seem to play a significant role in the prediction of burnout in the present study. Several suggestions for future studies, as well as implications for mental health practitioners, are discussed.

## 1. Introduction

There is a great interest in the issue of burnout among health and mental health professionals [1,2,3,4]. Burnout induces hopelessness, anger, and cynicism in human service professionals [5]. Burnout constitutes a stable, negative, work-related mental state of “normal” individuals, primarily characterized by exhaustion [6] and accompanied by stress, reduced effectiveness, and motivation, as well as the development of dysfunctional attitudes and behaviors at work [7]. It also entails depersonalization, cynicism, and a lack of personal achievement. The causes of burnout, in general, are related to the stressful factors in the work context and the attitude towards work [8]. Job demands have been associated with physical and psychological costs for healthcare professionals [9]. Therapists who have more experience seem to be less likely to cope with burnout compared to less experienced people [10]. Moreover, lack of supervision is a factor associated with burnout [11]. Burnout has also been related to vicarious trauma [12].

Vicarious trauma refers to the indirect trauma that may occur to people working with traumatized clients, such as health professionals, and includes nightmares, phobic thoughts, images that suddenly appear in the mind, suspicion of others’ intentions [13], decreased empathy [14], and, on a professional level, a loss of efficiency in providing therapeutic services while professional satisfaction decreases [15]. Therapists who experience vicarious trauma deal with difficulties regarding their own emotion regulation and their personal relationships [16]. Therapists who face vicarious trauma via their clients quite often find themselves experiencing or re-experiencing traumatic conditions on their own [17]. Taking the above into consideration, experiencing vicarious trauma has been found to be one of the highest work risks for therapists’ mental health [18]. It may even change therapists’ values and beliefs [19]. Pearlman and Saakvitne [20], as cited in Rauvola, Vega, and Lavigne [21], characterized it as the transformation of the therapists’ inner experience as a result of their empathic engagement with their clients and the traumatic material they carry. Vicarious trauma in therapists has been connected to women [22], little work experience [23], and a lack of supervision [24].

Empathy is the ability to understand and share the emotions, thoughts, and behavior of another person [25]. It helps the therapist understand the client’s internal frame of reference both accurately and by taking into account the emotional features associated with it as if it were their own, but forgetting the “as” [26]. By being empathetic, the therapist offers quality care and treatment and simultaneously is able to protect one’s personal mental health [27]. Empathy has been found to be higher in women in comparison to men and related to an interest in a counseling-type career [28].

The positive relationship between empathy and burnout has been documented by many studies [29,30]. This may be due to the professionals’ need to invest in the patients’ painful emotions, thus driving themselves to exhaustion [31]. The therapists’ compassion leads to depersonalization in order to protect themselves emotionally and spiritually. This depersonalization, however, does not bring relief to the therapists and reduces their satisfaction associated with offering help. Thus, the therapists become ineffective because of their inability to connect. Furthermore, their very willingness to remain in an empathic alliance with a client exposes the client to stress, which can lead to burnout [32]. Due to the lack of reciprocity in these relationships, professionals report feeling depleted of emotional resources, which leads to “emotional exhaustion.” Burnout is also related to vicarious trauma [12], since providing psychological assistance to people who have experienced trauma results in mental health professionals repeatedly encountering detailed narratives of traumatic events. In addition, they witness the emergence of strong emotional responses from their clients [18].

The literature review so far shows that, except for school counselors’ emotional exhaustion [33], there is limited research regarding burnout in Greek mental health professionals. To the best of our knowledge, no research has been conducted so far regarding vicarious trauma and empathy in relation to burnout among mental health professionals who practice psychotherapy in Greece.

Grounded in past research, we hypothesized that all the aforementioned variables would be significantly correlated. Similarly, the study’s goal was to investigate the relationship between demographic characteristics and them.

## 2. Materials and Methods

### 2.1. Participants

The recruitment of the sample was carried out by posting an announcement on a master’s degree student’s personal account as well as via specific groups on social media, such as Facebook. Thus, participants responded online to the anonymous survey questionnaires implemented via the Google Forms platform. Greek psychologists, psychiatrists, and other professionals that provided psychotherapy treatment to others were included in the sample. Participants who lived abroad, although they had Greek nationality, were excluded from the sample. Before completing the questionnaire, the participants had to sign a consent form that described the purpose of the research, the procedures to be followed, and the researcher’s obligation to comply with the code of ethics. The research was approved by the University of Thessaly’s Research and Ethics Committee because the majority of the work was done as part of an MSc thesis. Overall, 214 people took part, including 182 women (85%) and 32 men (15%). The highest percentages, regarding the demographics, were among mental health practitioners between 24 and 30 years of age (30.4%), psychologists and psychotherapists (43%), and professionals with graduate studies (64%). All demographic information is provided in Table 1.

### 2.2. Materials

The following self-report scales were used to collect the survey data: Participants reported their age, gender, marital status, specialty, years of experience, and years of supervision in an improvised demographics questionnaire.

The degree to which professionals experienced occupational burnout was measured by the counselor burnout inventory (CBI) [34], validated by Kounenou et al. [35], for which the participants had to state the frequency with which they encounter burnout on a 5-point scale (1 = “never” to 5 = “always”). It includes 20 items (e.g., “I feel exhausted due to my work as a counselor/therapist”). Cronbach’s alpha for the counselor burnout inventory in this study was 0.882. Indirect trauma was assessed by the vicarious trauma scale (VTS) [36], which contains 8 statements (e.g., “My job involved exposure to distressing materials and experiences”) ranging from 1 (“does not represent me at all”) to 7 (“represents me perfectly”). A higher score indicates greater levels of vicarious trauma [37]. Additionally, 3 questions applied to Adams, Matto, and Harrington’s [38] research on vicarious trauma were used, which the research subjects had to answer, marking their response on a 3-point scale where 1 corresponded to “not at all” and 3 to “very much.” Cronbach’s alpha for the vicarious trauma scale in this study was 0.780.

To assess empathy, participants were asked to respond to the Jefferson Scale of Empathy-Health Professionals Version (JSE-HP) [39], translated and validated by Tziala [40]. It is a reliable and valid self-report scale [41]. It consists of 20 statements (“Because people are different, it is almost impossible for me to see things from my patients’ perspectives”), in which the participants indicate their degree of agreement or disagreement on a 7-point Likert scale (1 = “strongly agree,” 7 = “strongly disagree”). Cronbach’s alpha for the Jefferson Scale of physician empathy (HP) in this study was 0.811.

### 2.3. Statistical Analysis

We checked for linearity and found no deviation. We tested for normality; the criteria for skewness and kurtosis were met, and the indexes did not exceed the absolute values of 3 and 10, respectively [42,43]. There were no missing values. The correlations between the measured variables, means, and standard deviations were calculated. Multiple regression analysis was performed to investigate the contribution of various factors that have been shown to be associated with burn-out [44], such as gender, marital status, years of supervision, years of work experience, vicarious trauma, and empathy. SPSS-21 was used for the above analyses.

## 3. Results

The strongest, statistically significant correlation was between burnout and vicarious trauma. We also came across non-significant correlations, such as those between vicarious trauma and empathy or years of supervision, as well as between empathy, years of supervision, and work experience. Means, standard deviations, and correlations among the psychological variables and the demographic factors are displayed in Table 2.

In addition, multiple regression analysis was used to test if demographic factors, vicarious trauma, and empathy significantly predicted burnout. Initially, it was found that neither gender (β = 0.001, ns) nor marital status (β = 0.077, ns) significantly predicted burnout. The slopes of the regression lines were not statistically significant (*F*(1, 212) = 0.00, ns and *F*(2, 211) = 0.62, ns, respectively). The demographic factors explained 0.06% of the burnout variance. In the next step, years of supervision significantly contributed to the burnout prediction (β = −0.186, *p* < 0.01), and the slope of the regression line was statistically significant, *F*(3, 210) = 2.952, *p* < 0.05. Years of supervision explained an additional 3.5% of the burnout variance. Next, years of work experience did not significantly predict burnout (β = −0.071, ns), *F*(4, 209) = 2.397, *p* = 0.051, whereas empathy significantly predicted burnout (β = 0.151, *p* < 0.05), *F*(5, 208) = 2.960, *p* < 0.05. These factors explained additional 0.3% and 2.3%, respectively, of the burnout variance (β = 0.592, *p* < 0.001), *F*(6, 207) = 23.280, *p* < 0.001. All the variables explained 40.3% of the burnout variance (Table 3).

Finally, we performed several mediation and moderation analyses, using vicarious trauma, empathy, years of supervision, and years of work experience as independent variables, mediators or moderators interchangeably, and burnout as a dependent variable, but we did not discover statistically significant results.

## 4. Discussion

The present study aimed to determine the nature of the connection between burnout, vicarious trauma, and empathy, as well as other demographic factors, among mental health practitioners practicing psychotherapy, on the grounds that research in this field is limited, especially in Greece. It revealed that these psychological variables have a positive and statistically significant relationship with respect to burnout on the one hand and vicarious trauma and empathy on the other.

Initially, gender and marital status were found to be unrelated to therapist burnout. This finding was consistent with previous studies [45,46], while others [47,48] have found women therapists to report a higher level of emotional exhaustion than male colleagues. Maslach and Jackson [49] have found a negative correlation between marriage and the occurrence of burnout.

Furthermore, burnout was strongly and positively related to vicarious trauma. This finding supports the findings of Rupert et al. [46], who discovered that uncontrolled and unresolved, chronic stress and psychological distress, symptoms that may be caused by indirect trauma, aggravated burnout symptoms using the jobs demands and resources model (JD-R) [50] and the conservation of resources (COR) model [51].

Burnout was positively related to empathy as well, but weakly. Emotional empathy may be the key to a therapist helping people, but it is also the skill that makes mental health professionals sensitive to the “cost of care” [31].

Moreover, it was found that therapists who have more years of supervision seem to be less likely to confront burnout, probably because during discussions with their colleagues they develop coping strategies to deal with stressors compared to therapists who have no supervision. The preventive role of supervision against burnout among mental health professionals has been documented by other researchers as well [52].

The current study has various limitations, such as the opportunity sample, the cross-sectional design, which makes it difficult to draw conclusions on causal relationships, and the heterogeneity of the group of mental health professionals with respect to their studies, their therapy training, their years of experience, and the patients they treat. For example, therapists working with personality disorders, particularly borderline personality disorder, may be more prone to experiencing burnout [53]. Moreover, this research was conducted in the midst of the COVID-19 pandemic, and this made access to mental health centers or other agencies difficult. However, our study has several strengths as well. First of all, it can lead to useful results concerning the treatment of traumatized clients. Furthermore, patients treated by burned-out therapists may have poorer outcomes than patients treated by non-burned-out therapists [54], and burned-out mental health professionals may become less involved in the therapeutic process, emotionally distant from their clients, and generally lower the quality of their work [46,49].

As for future directions, burnout can be associated with other factors related to vicarious trauma, such as workload, a controversial professional role, and authoritarian leadership that can make a therapist vulnerable [22]. Future research can also focus on the ways therapists can cope with burnout. Clinicians have been found to reduce their investment in the relationship with their clients as a form of self-protection [55]. It is this characteristic of burnout that Maslach [6] termed “depersonalization.” The findings of the current research indicate the importance of vicarious trauma and empathy in affecting mental health professionals’ burnout. This may contribute to significant developments in the field of solutions for therapists that deal with such difficulties.

## 5. Conclusions

It was discovered that vicarious trauma and years of supervision greatly influence the occurrence of burnout in Greek mental health professionals practicing psychotherapy. Empathy was also found to have an impact on therapists’ burnout. However, no other demographic factor was found to be predictive of burnout. These results confirm that vicarious trauma and empathy lead to therapists’ vulnerability, while supervision seems to provide a solution to their burnout symptoms.

Currently, this research continues with the addition of individual variables, such as mental health professionals’ personality and self-compassion, although it would be rather interesting to conduct the same research in various cultural settings.

## Figures and Tables

**Table 1 ijerph-20-03503-t001:** Demographics.

Demographic Characteristics	*n* (%)
Age group, years	
24–30	65 (30.4)
31–36	47 (22.0)
37–42	48 (22.4)
43–48	25 (11.7)
49–54	18 (8.4)
55+	11 (5.1)
Gender	
Male	32 (15.0)
Female	182 (85.0)
Marital status	
Single	43 (20.1)
In a relationship	95 (44.4)
Married	76 (35.5)
Occupation-Specialty	
Psychologists	81 (38.0)
Psychiatrists	6 (2.8)
Psychologists-Psychotherapists	92 (43.0)
Psychiatrists-Psychotherapists	11 (5.1)
Psychotherapists, unrelated to psychology and psychiatry	24 (11.2)
Studies	
Private college/Public university undergraduate	61 (28.5)
Graduate	137 (64.0)
PhD	16 (7.5)

**Table 2 ijerph-20-03503-t002:** Correlations between measured variables.

Variables	Mean	SD	CBI	VTS	JSE-HP	Years of Supervision	Years of Work Experience
CBI	2.01	0.57	1				
VTS	2.28	0.54	0.60 ***	1			
JSE-HP	3.90	0.36	0.16 *	0.08 ns	1		
Years of supervision	2.95	0.925	−0.19 **	−0.02 ns	−0.04 ns	1	
Years of work experience	2.22	1.487	−0.15 *	−0.16 **	−0.01 ns	0.51 ***	1

*Note*. * *p* < 0.05, ** *p* < 0.01, *** *p* < 0.001, ns: non-significant, CBI: counselor burnout inventory; VTS: vicarious trauma scale; JSE-HP: Jefferson scale of empathy-health professionals version.

**Table 3 ijerph-20-03503-t003:** Prediction of burnout by demographic and psychological measures.

	Regression Coefficients		
	Unstandardized	Standardized	
	Β	β	*t*
Step 1 (Δ*R*^2^ = 0.000) Gender	0.001	0.001	0.011 ns
Step 2 (Δ*R*^2^ = 0.006) Marital status	0.040	0.077	1.116 ns
Step 3 (Δ*R*^2^ = 0.040) Years of supervision	−0.115	−0.186	−2.752 **
Step 4 (Δ*R*^2^ = 0.044) Years of work experience	−0.027	−0.071	−0.863 ns
Step 5 (Δ*R*^2^ = 0.066) Empathy	0.240	0.151	2.242 *
Step 6 (Δ*R*^2^ = 0.403) Vicarious trauma	0.624	0.592	10.80 ***

*Note*. * *p* < 0.05, ** *p* < 0.01, *** *p* < 0.001. ns: nonsignificant.

## Data Availability

All data generated or analyzed during this study are included in this published article.

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
