# Peer review of "The Interplay among Empathy, Vicarious Trauma, and Burnout in Greek Mental Health Practitioners"

_ijerph, 2023, doi:10.3390/ijerph20043503_

Round 1

Reviewer 1 Report

Thank you for your interesting submission. There are just a few suggestions for revisions as below:

- Please have a native English speaking/writing individual review and revise your manuscript. There are several missing articles (i.e., "a" and "then), as well as phrases/sentences that don't make sense to me (e.g., page 1, line 54 - "confront burnout", page 2, line 54 "not rarely therapists", page 2, line 68 "joined with women" etc.).

- On page 1 line 35, when you state "there has been, recently, a growing interest in the issue of burnout among health and mental health professionals" - your citations are a bit too dated to make such a statement (most are 10+ years old). Please find citations from the past several years, if you would like to make such a statement.

- The paragraph starting on page 2, line 92 should be in the methodology section.

- Please provide some details on how you know respondents are from Greece. Given that you posted the study on Facebook, how do you know you did not receive responses from individuals outside of Greece (unless you had a question in the survey, in which case please make note of this in the methodology section, since you state at the beginning of the paper that this study is to explore burnout among Greek practitioners).

- On page 3, line 112, please include percentages in parentheses, when providing absolute numbers on gender.

- In the table, please add ".0" to be consistent in the number of decimal points used for all numbers.

- Please explain if data needed to be cleaned, and how missing data was handled.

- Please provide a clearer demographic explanation to "In relation with/without" - since I do not understand what is meant by this phrase.

- In the Conclusion section, I wasn't clear about the need to include the "(especially emotional one)" after empathy. Please provide a clearer phrase or explanation, since this did not make sense to me.

Thank you again for sharing your manuscript with me.

Reviewer 2 Report

The article sounds like a very good initial paper on an important subject. Although data are. well described the initial sample is small and the results are in line with sample size. One can expect that this research can be replicate in other countries and with bigger sample to enlarge knowledge. Furthermore,  it needs a better conceptualisation on core concepts like empathy that still looks a very open concept (I suggest to find a scale for those kind of concepts).
